# Amplicon sequencing provides more accurate microbiome information in healthy children compared to culturing

Shashank Gupta[1,5], Martin S. Mortensen [1,5], Susanne Schjørring[2], Urvish Trivedi[1], Gisle Vestergaard[1], Jakob Stokholm [3], Hans Bisgaard [3], Karen A. Krogfelt [2,4] & Søren J. Sørensen [1]

Next-Generation Sequencing (NGS) of 16S rRNA gene is now one of the most widely used application to investigate the microbiota at any given body site in research. Since NGS is more sensitive than traditional culture methods (TCMs), many studies have argued for them to replace TCMs. However, are we really ready for this transition? Here we compare the diagnostic efficiency of the two methods using a large number of samples ($n = 1,748$ fecal and $n = 1,790$ hypopharyngeal), among healthy children at different time points. Here we show that bacteria identified by NGS represented 75.70% of the unique bacterial species cultured in each sample, while TCM only identified 23.86% of the bacterial species found by amplicon sequencing. We discuss the pros and cons of both methods and provide perspective on how NGS can be implemented effectively in clinical settings.

[1] Section of Microbiology, Department of Biology, University of Copenhagen, 2100 Copenhagen, Denmark. [2] Department of Bacterial, Parasites and Fungi, Statens Serum Institut, 2300 Copenhagen S, Denmark. [3] Copenhagen Prospective Studies on Asthma in Childhood, Faculty of Health Sciences, University of Copenhagen, Copenhagen University Hospital Gentofte, Copenhagen, Denmark. [4] Virus and Microbiological diagnostics, Statens Serum Institut, 2300 Copenhagen S, Denmark. [5]These authors contributed equally: Shashank Gupta, Martin S. Mortensen. Correspondence and requests for materials should be addressed to K.A.K. (email: kak@ssi.dk) or to S.J.S. (email: sjs@bio.ku.dk)

For decades, microbiology has relied on traditional culture methods (TCMs) to identify bacteria involved in human health or disease: typically, samples are sent to a laboratory, and bacteria from these samples are enriched in broth and/or agar-based media, isolated and biochemically characterized for identification. TCMs enable classification of bacteria to species and strain level, and in a clinical setting, the right choice of media and incubation settings will enable selective growth of certain pathogenic bacteria, the possibility to evaluate growth rate and antimicrobial susceptibility testing. However, TCMs have many shortcomings: a bias towards bacteria that readily proliferate under laboratory conditions[1,2], require extended incubation time to identify organisms present[3,4], only provides qualitative results, and identifies only a fraction of the diverse microbial populations in question[5]. TCMs continue to be the mainstay for clinical laboratories while NGS will potentially replace this in the future.

Molecular diagnostic techniques, such as Polymerase chain reaction (PCR), DNA fingerprinting and Next-Generation Sequencing (NGS) have been game changers[2,6]. These techniques have become increasingly rapid, sensitive and cost-efficient[7,8]. For NGS of 16S rRNA gene, DNA is first extracted, a specific region of 16S rRNA gene is amplified, sequenced, and then identification of generated sequences is based on similarity to reference 16S rRNA gene sequences available in public databases. The major advantages of 16S rRNA gene NGS are that the method does not rely on whether or not the bacteria in a sample is culturable, the relative abundance of all bacteria in the sample can be determined, and it allows for parallel sequencing of hundreds of samples simultaneously and results can be obtained on the same day as the sample was taken[9]. Due to an increase in the number of PCR and DNA sequencing facilities available, the use of 16S rRNA gene sequencing is not limited to research purposes, but may soon be utilized by clinical laboratories[10]. However, 16S rRNA gene amplicon sequencing also has limitations: the primers used for amplification will introduce a bias as they bind to regions that are not 100% conserved across all bacteria, and, in most cases, bacteria can only be identified to genus level due to high similarity between 16S rRNA gene from closely related species, furthermore this method does not provide information on antimicrobial susceptibility.

Efforts to compare the results from culturing and amplicon sequencing have primarily focused on the identification of pure cultures. Woo et al.[10] made a thorough review of the usefulness of 16S rRNA gene sequencing, where they listed diseases for which the causative bacteria cannot be cultured. Comparison between the methods with respect to bacterial communities has been performed by only a few studies, looking at the bioburden in chronic wounds[5] or on a very small number of subjects[11,12].

To thoroughly evaluate whether NGS is ready to replace TCMs in diagnostics, we collected fecal samples and hypopharyngeal aspirations from children part of the Copenhagen Prospective Studies on Asthma in Childhood$_{2010}$ (COPSAC$_{2010}$) cohort[13]. Initially, the microbiota of all samples were characterized using TCMs, then followed by NGS of 16S rRNA gene; allowing us to compare 3,538 samples that have both been cultured and sequenced. Based on the results obtained by comparing TCMs with amplicon sequencing, using state-of-the-art bioinformatics and statistics, there were some discordances; TCMs identified up to 8 bacterial species per sample, whereas NGS identified up to 140 unique species per sample. We show that bacteria identified by NGS represented 75.70% of the unique bacterial species cultured in each sample, while TCMs only identified 23.86% of the bacterial species found by amplicon sequencing. Our data suggest that while NGS is better at describing the microbiome in samples from healthy children, TCMs still offers some additional information that is needed in a clinical setting. The workflows and data analysis used in this study have limitations in a clinical setting, but with the development of a standardized and automated pipeline sequencing should be ready to replace TCMs.

## Results

**Sample composition by culturing.** We cultivated bacteria from 1,790 hypopharyngeal samples from children aged 1 week ($n = 537$), 1 month ($n = 626$) and 3 months ($n = 627$) and 1748 fecal samples from the same children aged 1 week ($n = 542$), 1 month ($n = 597$) and 1 year ($n = 609$). From the totality of 3,538 samples, we isolated 9,832 pure cultures. The fecal samples contained on average 2.7 bacterial species, most commonly we observed *Escherichia coli* (56.59%), *Staphylococcus epidermidis* (29.46%), and *Enterococcus faecalis* (23.20%) among all samples. Similarly, the hypopharyngeal samples contained on average 2.4 bacterial species, most commonly *Staphylococcus aureus* (48.28%), *Staphylococcus epidermidis* (41.54%), and *Corynebacterium spp.* (28.42%) (Table 1; full list in Supplementary Table 1).

**Sample composition by next-generation sequencing.** A total of 195,078,994 high-quality paired sequences were obtained from the same 3,538 samples. The samples were rarefed to 1,806 reads per sample, which represented 8,161 amplicon sequence variants (ASVs). We classified the ASVs to species level (4% could be identified), except for a few families, e.g. *Enterobacteriaceae*, where members of different genera were not distinguishable. Several genera had multiple ASVs associated with them, but it was not possible to determine which unique species they represented. The majority belonged to the phyla Bacteroidetes, Firmicutes, Proteobacteria, Actinobacteria, Verrucomicrobia, and Fusobacteria, which corresponded to 99.80% of all sequencing reads (Supplementary Table 2, Supplementary Fig. 1). Bacteroidetes (33.27%) and Firmicutes (26.59%) accounted for the major phyla in fecal samples, whereas it was Firmicutes (61.9%) and Proteobacteria (29.78%) in hypopharyngeal samples (Supplementary Table 2).

The dominant families, average abundance >10%, in fecal samples were *Bacteroidaceae* (29.11%), *Enterobacteriaceae* (23.86%) and *Bifidobacteriaceae* (15.29%), whereas, *Streptococcaceae* (25.90%), *Staphylococcaceae* (26.39%) and *Moraxellaceae* (15.85%) dominated hypopharyngeal samples (complete list in Supplementary Table 3).

**Table 1 The five bacteria species most identified by culturing from each sample type. Percentages are the percent of samples they have been found in**

| Top Five | Hypopharyngeal | | Fecal | |
|---|---|---|---|---|
| 1 | *Staphylococcus aureus* | 48.28% | *Escherichia coli* | 56.59% |
| 2 | *Staphylococcus epidermidis* | 41.54% | *Staphylococcus epidermidis* | 29.46% |
| 3 | *Corynebacterium spp* | 28.42% | *Enterococcus faecalis* | 23.20% |
| 4 | *Moraxella catarrhalis* | 20.59% | *Klebsiella pneumoniae* | 14.99% |
| 5 | *Streptococcus mitis/oralis* | 20.06% | *Staphylococcus aureus* | 12.71% |

**Direct comparison of sample composition**. To establish how well the culturing and NGS data correlated, we calculated what proportion of the sequencing reads, within each sample, which represented bacteria cultured from that sample. The proportion of sequencing reads matched to cultured bacteria were highest at phylum level and decreased as we moved towards species level. Furthermore, within each taxonomic level, the proportion of matched sequencing reads were highest at 1 week and decreased over time. Moreover, there were large differences between the sample types; bacteria cultured from fecal samples were less abundant (mean = 21.38%) than hypopharyngeal samples (mean = 49.65%) (Fig. 1).

**Comparison using closed reference taxonomical assignment**. To improve the resolution of the amplicon sequencing data and allow for comparison at species level we performed closed-reference OTU picking at 100% identity for all ASVs. We compiled a reference database of type strains, from the Ribosomal Database Project (RDP) database, for the species identified by culturing. Isolates not identified to species level were disregarded. We identified 167 unique cultured bacteria, 22 were only identified to genus level (Supplementary Table 4). In addition, we pooled the bacteria that had identical V4 sequences, giving us 106 unique bacterial species and groups to compare (Supplementary Table 4). Among these 106 unique bacterial species, 40 genera were found in total, out of which the 5 most abundant genera

were *Staphylococcus*, *Escherichia/Shigella*, *Enterococcus*, *Moraxella*, and *Streptococcus* (Supplementary Table 5).

There were large differences in the sensitivity of the two methods, TCMs identified no more than 8 bacterial species per sample, with average 2.3 at 1 week, 2.19 at 1 month, and 2.22 at 1 year in fecal sample and 2.41 at 1 week, 2.42 at 1 month, and 2.42 at 3 months in hypopharyngeal samples. In comparison, NGS identified up to 140 unique species per sample, averaging 22.55 at 1 week, 21.94 at 1 month, and 52.22 at 1 year in fecal samples and 16.12 at 1 week, 20.12 at 1 month, and 25.18 at 3 months in hypopharyngeal samples (Table 2).

We then evaluated how many of the 106 identified unique species from the TCMs had a matching NGS sequence. Hypopharyngeal samples had the highest proportion of matching sequences (76.63%), while a low percentage of the fecal samples matched sequences in our reference database (27.63%) (Supplementary Fig. 2). Moreover, as the infant gets older, the mean proportion of matching sequences decreased. We observed that 1 week fecal samples had the highest percentage (40.36%) matched to reference database compare to 1 month (35.86%) and 1 year (8.21%). Similarly, among hypopharyngeal samples, 1 week had the highest percentage (80.67%) matched to reference database compare to 1 month (76.03%), and 3 months (73.71%). This could resemble the increasing abundance of anaerobic bacteria in the samples with time, which is especially found in the fecal samples.

When considering a presence/absence scenario for the 106 species, 75.7% of the times a bacterium was found and

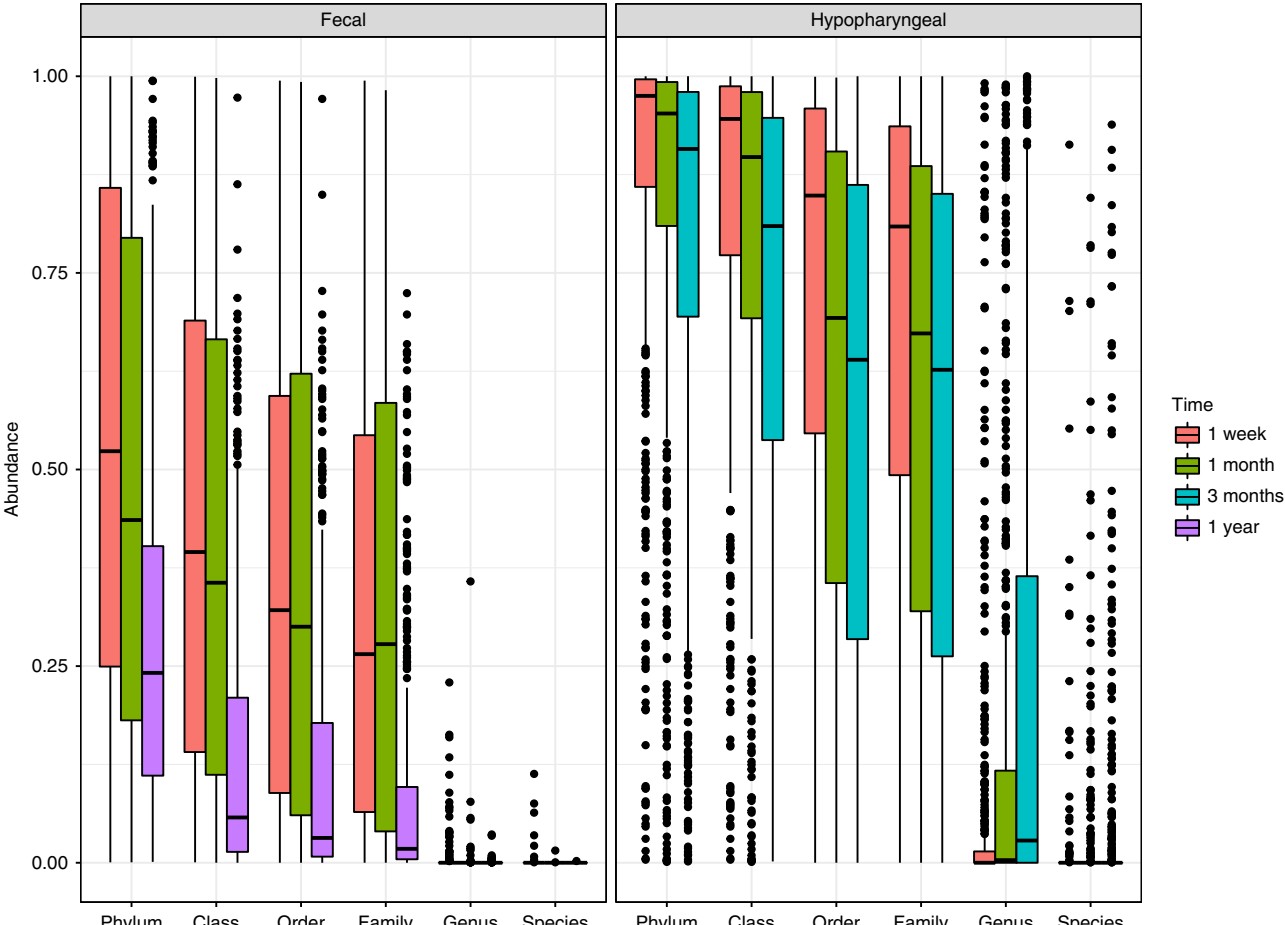

**Fig. 1** Boxplot showing the mean relative abundance of bacteria, classified using ASVs, matching bacteria identified, by culturing, in each sample. The analysis was performed for fecal and hypopharyngeal samples separately and at all taxonomic levels from Phylum to Species (x-axis) and the color of each box denotes the timepoint

**Table 2 Richness of samples by closed-reference OTU picking and culturing. Shown for all samples and split by sample type and sample time-point. The mean, standard deviation (SD), minimum number of species (Min), and the maximum number of species (Max) identified using both methods are listed**

|  | Type | Time | Mean | Min | Max | SD |
|---|---|---|---|---|---|---|
| *Culture* | Fecal | One Week | 2.30 | 0 | 5 | 1.00 |
|  |  | One Month | 2.19 | 0 | 6 | 1.03 |
|  |  | One Year | 2.22 | 0 | 7 | 1.08 |
|  | Hypopharyngeal | One Week | 2.41 | 0 | 7 | 1.20 |
|  |  | One Month | 2.42 | 0 | 6 | 1.23 |
|  |  | Three Months | 2.42 | 0 | 8 | 1.26 |
| *Amplicon sequencing* | Fecal | One Week | 22.55 | 7 | 82 | 10.51 |
|  |  | One Month | 21.94 | 3 | 100 | 9.27 |
|  |  | One Year | 52.22 | 8 | 119 | 18.00 |
|  | Hypopharyngeal | One Week | 16.12 | 2 | 137 | 9.02 |
|  |  | One Month | 20.12 | 3 | 140 | 10.77 |
|  |  | Three Months | 25.18 | 1 | 99 | 11.12 |

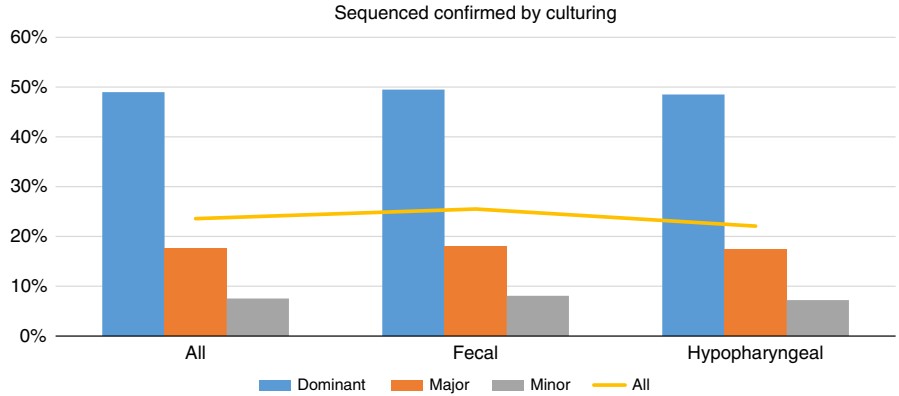

**Fig. 2** Barplot showing the percentage of bacteria identified by amplicon sequencing that were also identified by TCM in the same sample. The three groups of bars (*x*-axis) show the result for the different sample types (all together or fecal and hypopharyngeal samples separately). Blue bars show the result for dominant bacteria (abundance >10%), orange show major bacteria (1–10%), and gray show minor bacteria (<1%), while the yellow line indicates the overall percentage for each sample type

identified by culturing, amplicon sequencing identified the same bacteria from the same sample. Likewise, counting each time one of the 106 bacteria was identified in a sample, culturing had only identified the same bacterium from the same sample 23.86% of the times (Supplementary Table 6).

Some bacteria were only detected by culturing and not by amplicon sequencing. For example, culturing identified *Haemophilus influenzae* in 110 samples where it was not detected by sequencing, *Streptococcus pluranimalium* (*n* = 110), *Enterococcus casseliflavus* (*n* = 84), *Staphylococcus haemolyticus* (*n* = 53), and *Enterococcus faecium* (*n* = 43), none of which were identified by sequencing (Supplementary Table 7). NGS identified many of the taxa more frequently than culturing, *Haemophilus haemolyticus* was, except for one culture positive sample, only detected by sequencing (*n* = 1,231). Similarly, e.g., *Streptococcus salivarius* and *S. vestibularis* (sequencing: *n* = 1,997, culturing: *n* = 16, both: *n* = 133), *Streptococcus mitis* (sequencing: *n* = 1,781, culturing: *n* = 16, both: *n* = 359).

The relative abundance in the sequenced results within each sample affected the likelihood of detecting the bacteria by culturing. When classifying bacteria, in each sample, by their relative abundances (by sequencing) higher than 10 % as dominant, from 1 to 10% as major, and lower than 1% as minor, we found a positive correlation (chi-square test, p-value < 0.05) to the probability of identifying it by culturing. Of the dominant bacteria, 49.76% were also identified by culturing, while 17.88% of

major and 7.66% of minor bacteria were identified (Fig. 2, Supplementary Table 6). This correlation were very clear when looking at the probability of identifying specific bacteria such as, *Staphylococcus* GroupA (92.69, 67.80, or 42.88% when dominant, major or minor, respectively), *Escherichia/Shigella* group (79.23, 31.16, or 6.79%, respectively) or *Enterobacteriaceae* group A (69.52, 33.33, or 10.44%, respectively) (Supplementary Table 7).

**Amplicon sequencing resolution**. Amplicon sequencing was more sensitive than culturing, identifying more bacteria per sample than culturing (mean 26.4 vs 2.33 bacteria per sample, Table 2), despite including only sequences with 100% matches to the reference database, but did not have sufficient resolution. In a clinical setting, the difference between the species *S. aureus* and *S. epidermidis* is very relevant, but the V4 region of the 16S rRNA gene from the two species are 100% identical. For *Enterobacteriaceae*, the problem is more pronounced, as many genera cannot be separated based on the sequence of their V4 region of the 16S rRNA gene. An *in silico* comparison of the resolution, if both variable region V3 and V4 had been sequenced, found that for species from *S. aureus* group, three would still have identical sequences (*S. epidermidis*, *S. capatis*, and *S. caprae*), but notably, *S. aureus* would not be identical to any other species (Fig. 3a). For *Enterobacteriaceae*, all species could be further separated if variable region 3 were included (Fig. 3b).

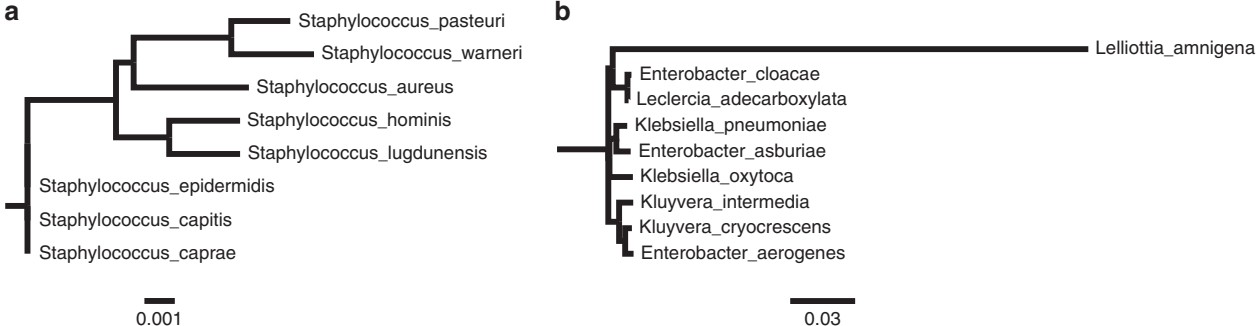

**Fig. 3** Phylogenetic trees showing how groups of bacteria with identical V4 sequences separate when using the V3–V4 regions. **a** *Staphylococcus aureus* group and **b** *Enterobacteriaceae* group A

Additionally, as shown earlier *H. influenzae* was found in 110 samples using TCM, but never using amplicon sequencing, while the closely related species *H. haemolyticus* was only found in a single sample using TCM, but in 1,231 samples by amplicon sequencing. The sequences for the two species are >99% identical (251/253 bp) and of the 110 samples where TCM had identified *H. influenzae* amplicon sequencing found *H. haemolyticus* in 92, and in none of these were *H. haemolyticus* found by TCM (110/92/0). For the other species that were often not identified by amplicon sequencing, we were not able to find a similarly clear pattern: The closest related species to *S. pluranimalum* (*S. thoraltensis*) were not found by sequencing either. *E. faecalis/durans/hirae* were the closest sequence to both *E. casseliflavus* and *E. faecium*, but when comparing the times they were found by TCM to when *E. faecalis/durans/hirae* were identified by amplicon sequencing and by TCM as well, the pattern were not as clear (84/27/7 for *E. casseliflavus* and 43/24/5 for *E. faecium*). For *S. haemolyticus* closest sequence were the *Staphylococcus GroupA* which were abundant by itself (53/41/36).

## Discussion

We performed a comparative evaluation for a large set of samples by means of culture-dependent and molecular diagnostics methods. Using a Naive Bayes classifier for taxonomical assignment of the sequencing data we were able to assign taxonomy to all ASVs, but only 4% of all sequencing reads were identified at species level. Some of the taxonomical unresolved ASV are likely to represent species that are not represented in RDP, while a large part will be from very well described bacterial genera or families where the V4 region of 16S rRNA gene does not provide sufficient resolution to differentiate multiple species. In addition to a decreased proportion of sequencing reads matching taxa cultured from each sample with each taxonomic level, we found a decrease over time and a lower proportion in fecal samples compared to hypopharyngeal samples. We speculate, that the two main reasons for this lower proportion in fecal compared to hypopharyngeal samples may in part be explained by the fact that no strict anaerobe bacteria were identified to species level using TCMs from the airways, and that the complex bacterial community of the gut contains species with very narrow growth requirements and slow growth rates, which are identified by amplicon sequencing but cannot be cultured using TCMs.

We used a naive Bayes classifier for our initial taxonomical classification of sequences, which ensures that all reads get the best possible taxonomical classification at the expense of some resolution. For many forms of research, this is important, as it enables classification of unknown bacteria without skewing the data towards well described bacterial taxa. In most clinical setups, the presence of a specific bacteria would be more important than

an unbiased overview with unknown bacteria. Therefore, we created a non-redundant database containing strain type sequences for all bacteria that were cultured from these samples, using TCMs, and performed closed reference taxonomical classification against it. While still being limited by the lack of cultured anaerobe bacteria and bacterial species not having a sequenced type strain, this approach closely resembles a likely clinical setup. With this approach, we matched 76.63% of sequences from hypopharyngeal samples to species cultured in the same sample and 26.50% from fecal samples (Supplementary Fig. 2). While this is an improvement compared to the initial analysis we still see that the upper respiratory tract harbors fewer uncultured bacteria in comparison to the gut where they represent a large percentage of the microbiota[14].

We found large differences in sensitivity of the two methods, identifying 7–20 times more unique species with NGS than with TCMs. Especially from fecal samples TCMs identify very few unique species, even slightly less than in hypopharyngeal samples and only NGS showed the expected increase in diversity that have been shown previous studies[15,16]. While NGS would be expected to have higher sensitivity, this comparison includes the data from closed reference tax assignment, which means that all bacteria identified by amplicon sequencing had been identified in at least once by TCMs, this cannot be attributed to an inability to culture any specific bacteria. To investigate the possibility of this being related to low abundant bacteria not being identified using TCMs, we grouped the bacteria sequenced in each sample as dominant (>10%), major (1–10%) or minor (<1%). Our comparison clearly shows that dominant bacteria, in general, were more likely to be identified by TCMs, and when comparing within individual species, this provided even stronger support for TCMs being biased by the relative abundance of each bacteria, which may be a good thing in a clinical setting, where you have no intention on initializing treatment on a bacteria that is not infecting but only colonizes. The surprisingly low number of fecal bacteria identified by TCMs may be due to more difficulty culturing at lower relative abundance. This was especially clear in the 1 year fecal samples, where <9% of the microbial community belonged to bacteria which were cultured (Supplementary Fig. 2). TCMs failed to detect ~50% of the dominant bacteria identified by amplicon sequencing, while >80% of the major bacteria were not detected by TCMs.

Sequencing just one variable region of 16S rRNA gene was not sufficient to consistently separate the bacterial species, identified by culturing, to species level. Our *in silico* analysis, extending the amplified region to include both variable region V3 and V4, showed an increased resolution when comparing clusters of species with 100% sequence identity in variable region V4. Additionally, our results show that amplicon sequencing had some problems with correct separation of sequences that differ in

as little as 1–2 bp out of 253 bp. An important step in the process is to create an error model that can infer if a single nucleotide difference is a sequencing error or an actual sequence variant, and it is possible that with the high sequence similarity the error model wrongly classifies the differences as a sequencing error and then changes the sequences to match the closest matching sequence.

With a well-curated reference database, for clinical relevant bacteria, a 16S rRNA gene amplicon sequencing workflow could be implemented to provide a plug and play output showing an overall picture of the microbial community and accurately identify relevant bacterial species. This combined with optimized oligonucleotide arrays for detecting various genes, including those encoding for resistance and toxins, as well as distinguishing specific species could very well be a game changer for diagnosticians. Our findings show that with the development of a standardized and automated pipeline sequencing will be ready to replace TCMs. However, even if TCM were to be replaced for clinical testing, there is still a need for further refining culture techniques for research involving bacterial behavior and interactions.

## Methods

**Ethics**. The study follows the principles of the Declaration of Helsinki and was approved by the Ethics Committee for Copenhagen (The Danish National Committee on Health Research Ethics) (H-B- 2008–093) and the Danish Data Protection Agency (2008–41–2599). Written informed consent was obtained from all participants. The study is reported in accordance with the STROBE guidelines[17]. Written consent for publication has been obtained from the parents or legal guardians of all participants.

**Study population**. The novel Copenhagen Prospective Study on Asthma in Childhood 2010 (COPSAC2010) is an ongoing Danish cohort study of 743 unselected pregnant women and their children followed prospectively from pregnancy week 24 in a protocol largely similar to the first COPSAC birth cohort (COPSAC2000)[13,18,19]. Recruitment lasted during 2009–10. Exclusion criteria were chronic cardiac, endocrinological, nephrological or lung disease other than asthma.

**Sample collection**. Hypopharyngeal aspirates were collected at 1 week (537 samples), 1 month (626 samples), and 3 months (627 samples) after birth. Fecal samples were collected at 1 week (542 samples), 1 month (597 samples), and 12 months (609 samples)[13]. All 3,538 samples were transported to Statens Serum Institut (Copenhagen, Denmark) where they were cultured and stored within 24 h of sampling.

**Culturing**. Bacterial samples were cultured with standard methods on non-selective and selective media (SSI Diagnostica, Hillerød, Denmark). One set of blood agar plates and chocolate agar plates (both supplemented with 5% horse blood) were incubated aerobically at 37 °C for 18–20 h. Another set of blood agar and chocolate agar plates were incubated under microaerophilic conditions (5% $CO_2$, 3% $H_2$, 5% $O_2$ and 87% $N_2$) at 37 °C for 48 h. Fecal samples were cultured on an anaerobic plate, under anaerobic conditions at 37 °C for 72 h. Subsequently, based on the growth on selective media, characteristics of colonies, and cellular morphology, all unique bacterial colonies were isolated. All bacterial isolates were identified biochemically by the automated identification system VITEK-2 (BioMérieux, France). Bacteria cultured anaerobically were not identified further. All isolates were preserved at −80 °C. No quantification was performed[20].

**Amplicon sequencing**. DNA extraction and 16S rRNA gene amplicon sequencing was done in the same way as published in our earlier studies[21]. Primers were removed from the MiSeq generated FASTQ files by cutadapt[22]. Further, reads were analyzed by QIIME2[23] pipeline using DADA2[24] to create sequencing error profiles, trim (first 8 bp of each read), truncate (forward reads to 180 bp, reverse reads to 160 bp), assemble read pairs, remove chimeras, infer the amplicon sequence variants (ASVs) present and assign taxonomy using a pre-trained Naive Bayes classifier [Silva Ref NR 99 (release 132)][25]. Based on the rarefaction curves for observed richness and Shannon diversity index (Supplementary Fig. 3), samples with <2,000 reads were excluded from the analysis, as well as 3 samples with an unusually high richness, which were suspected to be technical artifacts. In total 3,538 samples (1,748 fecal and 1,790 hypopharyngeal) were included for the comparisons. To avoid the bias due to sampling depth, the OTU table was multiple rarefied to 1,806 high-quality sequences per sample (90% of the minimum sample reads) using the in-house function.

**Closed reference OTU picking**. Based on the cultured species, a reference database was created from the matching type strains in the RDP database for V4 region only[26]. ASVs were matched at 100% using pick_closed_reference_otus.py script in QIIME among 3,538 samples.

The abundances of the bacteria were calculated as the percent of all reads, including those, which did not match to the reference database. Based on their abundances the bacteria were classified as dominant bacteria (abundance >10%), major (1–10%), or minor (<1%).

**Reporting summary**. Further information on research design is available in the Nature Research Reporting Summary linked to this article.

## Data availability

The sequences were submitted in NCBI Sequence Read Archive (SRA) under the Bioproject ID PRJNA543007. The contingency table showing which bacteria were isolated from each sample has been attached as Supplementary data file. All other data is available from the corresponding author.

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

## Acknowledgements

We express our deepest gratitude to the children and families of the COPSAC$_{2010}$ cohort study for all their support and commitment. We acknowledge and appreciate the unique efforts of the COPSAC research team. We thank Karin Pinholt Vestberg and April Cockburn (Section for microbiology, University of Copenhagen) for the help and support with DNA extraction, construction of the 16S rRNA gene amplicon libraries, and sequencing.

## Author contributions

S.G. and M.S.M. are the main authors of this paper. M.S.M. performed the DNA extraction and sequencing. S.S. and M.S.M. culture isolation and identification. S.G., M.S.M. and G.V. performed the bioinformatics analysis. J.S. sampled the infants. S.G., M.S.M., U.T. and G.V. helped interpret the data. This project was conceived and designed by S.J.S., K.A.K. and H.B. All the authors have read, revised, and approved the manuscript.

## Additional information

**Competing interests:** The authors declare no competing interests.

