## [Peer Review File · Communications Biology]

Reviewers' comments:

Reviewer #1 (Remarks to the Author):

This manuscript touches a very interesting topic of whether 16S rRNA amplicon sequencing is ready or not to replace the culturing in clinical diagnosis using a relatively big sample size. The methodology is nothing novel and the discoveries are highly expected that sequencing will obtain much more species than culturing. However, the comparisons still have their value in terms of comparing these two technologies in the clinical application scenario. It is good to have some big sample size data to show how the two technologies differ in terms of identifying the bacterial species. Hence, I think it holds the value to be published.

Several minor comments:

1. The detailed table of which bacteria species are cultured from which sample should be listed in the supplementary tables in detail. The table is the resource and part of the major value of this study.
2. Due to the mature of NGS and the popularity of third generation sequencing (Pacbio and Nanopore), full-length 16S rRNA sequencing will be much more accurate to achieve species-level identification. And these technologies may have more possibility to replace culturing rather than sequencing a hypervariable region of 16S rRNA discussed in this manuscript. Moreover, for high biomass samples like faecal samples, shotgun metagenomics can achieve a higher resolution to strain level can be used. In considering this the method used in this study cannot represent sequencing. The scope of the title culturing vs sequencing is not accurate, it is more accurate of culturing vs 16S rRNA hypervariable sequencing. This part should be mentioned in the discussion.
3. For data sharing, the authors should deposit the raw sequences data to the public database and also the metadata associated with the samples.

Reviewer #2 (Remarks to the Author):

This study compares V4 sequencing versus traditional culture in a large cohort. The size of the study is intriguing, but the article is confusingly written and does not simply lay out methods and findings. The number of reads per specimen is incredibly low. The article could be substantially improved by including more descriptive legends on the Figures. More description is needed on how TCM was conducted as the comparator method especially in terms of how the bacterial isolates were chosen for Vitek characterization.

line 21-22 - unclear whether these percentages describe the percentage of all species of bacteria detected or is the percent abundance covered. Also need to say at what taxonomical category this was covered.

line 79 - why rarefy the samples to so few reads per sample?

more discussion of the TCM is required. was every cultured bacteria speciated? usually this is not done for fecal culture.

There needs to be more discussion as to why NGS would miss any of the bacteria detected by culture.

Figure 1 - unclear from legend what the time period is covered. Unclear what unit the Abundance is in.

Figure 2 - this figure is totally unclear to me. It is not clear what the denominator is on the Y axis. The legend is not written in complete sentences. Why not just say "Culture detected XX% of the total number of species found by sequence in XX sample".

Reviewer #3 (Remarks to the Author):

The study by Gupta et. al compares the diagnostic efficiency of traditional culture methods and 16s rRNA sequencing using a large number of fecal and hypopharyngeal samples in healthy infants. The study is remarkable for its design, which pairs each methodology in each metagenome and in its scope, analyzing over 3,500 samples. This investigation highlights the strengths and weaknesses of each approach conveying the influence of sampling site, subject age, and microbial abundance within a sample on the efficiency of each methodology. The study is well-written and organized in a way that lends to understanding its key concepts. Overall, this investigation provides an important frame of reference for those hoping to understand the link between longstanding cultivation methods and a future shifting toward clinical applications of next generation sequencing.

One aspect of the paper which does not seem fully settled is whether or not the authors have reached a definite conclusion on the central question: "Are we ready for this transition?" While the authors conclude that with development of a "16s rRNA gene amplicon sequencing workflow" and "optimized oligonucleotide arrays for detecting various genes" we might be ready for replacement of traditional culture methods, they end by stating that further refinement of techniques "involving bacterial behavior and interactions" is needed. This conclusion feels somewhat vague in the context of the extensive analysis they have undertaken.

For instance, the authors make an effort to establish that resolution to the species level is often not possible with sequencing of 16s rRNA gene amplicons (especially in cases when only the v4 region is analysed). They also point out that in this study some potentially pathogenic organisms, such as haemophilus influenzae, were cultured only via traditional methods and not recovered from sequencing reads in a large number of metagenomes, a finding which in the right clinic setting is of huge significance. Additionally, the authors make the claim that TCMs are biased toward relative abundance of a microbial population within a metagenome and they claim that this "may be a good thing in a clinical setting, where you have no intention on initializing treatment on a bacteria that is not infecting but only colonizes". However, the importance of these findings does not seem directly linked to a strong conclusion about the current state of methodology in clinical application.

One question that may linger in the minds of readers is how 16s rRNA compares to other methods of high-throughput sequencing. In particular, shotgun metagenomics might overcome the lack of species/strain resolution seen with 16s rRNA and would allow researchers to more fully understand the behaviour and interactions within the metagenomes being studied. At the same time shotgun data is complex, data analysis is time consuming and the cost per sample is much higher than 16s rRNA. While these limitations may make shotgun metagenomics less relevant for imminent clinical application, a brief reflection on this modality may help orient the reader.

As best I am able to tell from reviewing the manuscript it does not seem the data generated in this study has been made available with a public database. In general, data availability is essential for credibility and allows proper scrutiny of analyses by the reviewers and the community.

Aside from these concerns, I would like to thank the authors for their ambitious data analyses. This research provides an important touchstone at a critical time for cultivation techniques in the clinical

setting.

Minor issues:

Line 16: Do we have references for "many studies have argued for them to replace TCMs"? This is stated in abstract but not explored in the scientific justification. Perhaps this position is advocated in the papers cited regarding 16s rRNA vs. TCM (5, 9-11)?

Line 67-69: Are the same children sampled multiple times (1 week, then 1 month, etc) or is each sample from a unique individual? This may be irrelevant, but might be helpful to know if these samples were meant to be understood as time-series or independently?

Line 38: "Molecular diagnostic techniques, such as Polymerase chain reaction (PCR), DNA fingerprinting and Next-Generation Sequencing (NGS) may be a game changer" seems a bit vague for a lead sentence. As the thrust of the article relates to diagnostic capacity of 16s rRNA are there any citations that could be applied here to support this claim of being a game-changer?

Line 78/161: Is 4% species-resolution when using amplicon sequence variants (ASV) typical? If this is comparable to other studies a citation would be helpful, if not an explanation for possible sequencing error/contamination might also be needed.

Line 102/258: Authors should state that RDP stands for Ribosomal Database Project.

Line 125-127: Is it possible to clarify this statement: "When considering a presence/absence scenario among the 106 species, bacteria identified by both culturing and sequencing represented 75.70 % of all cultured bacteria, but only accounted for 23.86 % of sequenced bacteria " Is this to say: Of the 106 identified unique species from culture, 75.7% could also be recovered in sequencing reads with 16s rRNA methodology. While these 106 culturable species represented just 23.86% of the unique bacteria found via 16s rRNA?

Reviewer comments

Author's comments

Reviewer #1

This manuscript touches a very interesting topic of whether 16S rRNA amplicon sequencing is ready or not to replace the culturing in clinical diagnosis using a relatively big sample size. The methodology is nothing novel and the discoveries are highly expected that sequencing will obtain much more species than culturing. However, the comparisons still have their value in terms of comparing these two technologies in the clinical application scenario. It is good to have some big sample size data to show how the two technologies differ in terms of identifying the bacterial species. Hence, I think it holds the value to be published.

Q1: The detailed table of which bacteria species are cultured from which sample should be listed in the supplementary tables in detail. The table is the resource and part of the major value of this study.

A1: Thanks for your kind suggestions. We have now included a link to the data in the data availability section.

Q2: Due to the mature of NGS and the popularity of third generation sequencing (Pacbio and Nanopore), full-length 16S rRNA sequencing will be much more accurate to achieve species-level identification. And these technologies may have more possibility to replace culturing rather than sequencing a hypervariable region of 16S rRNA discussed in this manuscript. Moreover, for high biomass samples like faecal samples, shotgun metagenomics can achieve a higher resolution to strain level can be used. In considering this the method used in this study cannot represent sequencing. The scope of the title culturing vs sequencing is not accurate, it is more accurate of culturing vs 16S rRNA hypervariable sequencing. This part should be mentioned in the discussion.

A2: We agree with the reviewer therefore we changed the title to "Culture vs amplicon sequencing". We have further revised the manuscript to amplicon sequencing (line 55, 65, 102, 129, 155, 156, 194, 214, 226 and 277).

Q3: For data sharing, the authors should deposit the raw sequences data to the public database and also the metadata associated with the samples.

A3: We apologise for not having made the data available before submission. We have now included a data availability section and made both sequencing and culturing data available. The sequences were submitted in NCBI Sequence Read Archive (SRA) under the Bioproject ID PRJNA543007. As the NCBI data is not available publicly before publication you can download all fastq files from NCBI reviewer link- <https://dataview.ncbi.nlm.nih.gov/object/PRJNA543007?reviewer=6786803bb25a21ovsuasnddc>

Reviewer #2 (Remarks to the Author):

This study compares V4 sequencing versus traditional culture in a large cohort. The size of the study is intriguing, but the article is confusingly written and does not simply lay out methods and

findings. The number of reads were specimen is incredibly low. The article could be substantially improved by including more descriptive legends on the Figures. More description is needed on how TCM was conducted as the comparator method especially in terms of how the bacterial isolates were chosen for Vitek characterization.

Q4: line 21-22 - unclear whether these percentages describe the percentage of all species of bacteria detected or is the percent abundance covered. Also need to say at what taxonomical category this was covered.

A4: Thank you for bringing this to our attention. The percentage described here refers to the average percentage of unique bacterial species identified in a given sample by the two methods (abundances not considered).

Q5: line 79 - why rarefy the samples to so few reads per sample?

A5: We agree that it would have been better if we could include more reads, but our analysis is sensitive to sequencing depth, which makes rarefaction unavoidable. Based on the rarefaction curves we chose to exclude samples with less than 2000 reads and then, to avoid bias of having unrarefied samples we rarefied to 90% of the reads in the sample with the least reads (2006 reads).

Despite the level of reads per sample, we reach a resolution where one read represents less than 0.06% of the total abundance, and we are confident in our results as any increased sequencing depth can be expected to have amplified the results.

Q6: more discussion of the TCM is required. was every cultured bacteria speciated? usually this is not done for fecal culture.

A6: We are not certain we understand this query, does the reviewer wish that we expand the TCM part in the method section or in the result section?

While it might not be how others have done this, our aim was to identify as many bacteria as possible using the standard media and incubation conditions.

We have updated line 270-276 in the method section so that they now read- "Subsequently, based on the growth on selective media, characteristics of colonies, and cellular morphology, all unique bacterial colonies were isolated. All bacterial isolates were identified biochemically by the automated identification system VITEK-2 (BioMérieux, France). Bacteria cultured anaerobically were not identified further. All isolates were preserved at -80°C. No quantification was performed". Furthermore, we added a missed reference to Stokholm et al 2011 where the isolates from the cohort's vaginal samples were presented.

Q7: There needs to be more discussion as to why NGS would miss any of the bacteria detected by culture.

A7: We have been surprised by this as well and decided to look at the data again. Of the 110 times *H. influenzae* was identified by TCM sequencing identified *H. haemolyticus* 92 times (which was not cultured in those samples). As the difference between the sequence of the two bacteria is just 2/253 bp, this indicates that the bioinformatic pipeline cannot correctly

differentiate between the two species. We have made this comparison for the 5 species mentioned in line 136-140, and added this as an extra paragraph in the results section for amplicon sequencing resolution (line 167-179).

Additionally, we have also added the following text to the discussion (line 231-237):

“Additionally, our results show that amplicon sequencing had some problems with correct separation of sequences that differ in as little as 1-2 bp out of 253 bp. An important step in the process is to create an error model that can infer if a single nucleotide difference is a sequencing error or an actual sequence variant, and it is possible that with the high sequence similarity the error model wrongly classifies the differences as a sequencing error and then changes the sequences to match the closest matching sequence.”

Q8: Figure 1 - unclear from legend what the time period is covered. Unclear what unit the Abundance is in.

A8: To improve the clarity of the figure legend we have changed it to- “Boxplot showing the mean relative abundance of bacteria classified using ASVs, matching bacteria identified, by culturing, in each sample. The analysis was performed for fecal (left panel) and hypopharyngeal (right panel) samples separately and at all taxonomic levels from Phylum to Species (x-axis) and the color of each box denotes the timepoint.”

Q9: Figure 2 - this figure is totally unclear to me. It is not clear what the denominator is on the Y axis. The legend is not written in complete sentences. Why not just say "Culture detected XX% of the total number of species found by sequence in XX sample".

A9: To improve the clarity of the figure legend we have changed it to- “Fig. 2: Barplot showing the percentage of bacteria identified by amplicon sequencing that were also identified by TCM in the same sample. The three groups of bars (x-axis) show the result for the different sample types (all together or fecal and hypopharyngeal samples separately). Blue bars show the result for dominant bacteria (abundance > 10%), orange show major bacteria (1-10%), and grey show minor bacteria (< 1%), while the yellow line indicates the overall percentage for each sample type.”

Reviewer #3 (Remarks to the Author):

The study by Gupta et. al compares the diagnostic efficiency of traditional culture methods and 16s rRNA sequencing using a large number of fecal and hypopharyngeal samples in healthy infants. The study is remarkable for its design, which pairs each methodology in each metagenome and in its scope, analyzing over 3,500 samples. This investigation highlights the strengths and weaknesses of each approach conveying the influence of sampling site, subject age, and microbial abundance within a sample on the efficiency of each methodology. The study is well-written and organized in a way that lends to understanding its key concepts. Overall, this investigation provides an important frame of reference for those hoping to

understand the link between longstanding cultivation methods and a future shifting toward clinical applications of next generation sequencing.

Q10: One aspect of the paper which does not seem fully settled is whether or not the authors have reached a definite conclusion on the central question: “Are we ready for this transition?” While the authors conclude that with development of a “16s rRNA gene amplicon sequencing workflow” and “optimized oligonucleotide arrays for detecting various genes” we might be ready for replacement of traditional culture methods, they end by stating that further refinement of techniques “involving bacterial behavior and interactions” is needed. This conclusion feels somewhat vague in the context of the extensive analysis they have undertaken.

A10: At this point we cannot conclude that we are ready for a transition away from traditional culture methods for clinical work. We believe that our results, together with the latest developments in sequencing technology, shows that the technological tools are available to make the transition. Unfortunately, we need a polished protocol/package that can rapidly handle, interpret and present sequencing result before this transition can be made. While we would prefer to provide a very strong conclusion about our results we find that this is not supported by the results presented in the manuscript.

The point we want to make with the last sentence is that while traditional culture methods might disappear from clinical laboratories, the techniques can still be improved as they are still highly relevant for research involving bacterial behavior and interactions. We have expanded the last sentence (line 244-245) so it now reads: “However, even if TCM were to be replaced for clinical testing, there is still a need for further refining culture techniques for research involving bacterial behavior and interactions. ”

Q11: For instance, the authors make an effort to establish that resolution to the species level is often not possible with sequencing of 16s rRNA gene amplicons (especially in cases when only the v4 region is analysed). They also point out that in this study some potentially pathogenic organisms, such as haemophilus influenzae, were cultured only via traditional methods and not recovered from sequencing reads in a large number of metagenomes, a finding which in the right clinic setting is of huge significance.

A11: This point is very relevant and was also raised by reviewer number 2. Therefore, we will refer to our answer to query number 7.

Q12: Additionally, the authors make the claim that TCMs are biased toward relative abundance of a microbial population within a metagenome and they claim that this “may be a good thing in a clinical setting, where you have no intention on initializing treatment on a bacteria that is not infecting but only colonizes”. However, the importance of these findings does not seem directly linked to a strong conclusion about the current state of methodology in clinical application.

A11: We agree that the references section is not linked to a strong conclusion, but this part is more speculative and we are of the opinion that our data do provide the necessary support to make a strong conclusion about the current state of methodology in clinical applications.

Q12: One question that may linger in the minds of readers is how 16s rRNA compares to other methods of high-throughput sequencing. In particular, shotgun metagenomics might overcome the lack of species/strain resolution seen with 16s rRNA and would allow researchers to more fully understand the behaviour and interactions within the metagenomes being studied. At the same time shotgun data is complex, data analysis is time consuming and the cost per sample is much higher than 16s rRNA. While these limitations may make shotgun metagenomics less relevant for imminent clinical application, a brief reflection on this modality may help orient the reader.

A12: Thanks for your kind suggestions. While we fully agree with this query, and have considered to discuss this in the manuscript, we felt that this would be a discussion for another article where the methods could be compared as well. Furthermore, as the reviewer mentions in the end, shotgun sequencing is less relevant for imminent clinical application. In response to reviewer 1 (Q2), bringing up that we did not clearly state the type of sequencing used in this study, we have changed the title of the manuscript to “Culture vs amplicon sequencing” for clarity. Furthermore, we have further revised the manuscript to amplicon sequencing (line 55, 65, 102, 129, 155, 156, 194, 214, 226 and 277).

Q13: As best I am able to tell from reviewing the manuscript it does not seem the data generated in this study has been made available with a public database. In general, data availability is essential for credibility and allows proper scrutiny of analyses by the reviewers and the community.

A13: We apologise for not having made the data available before submission. We have now included a data availability section and made both sequencing and culturing data available. The sequences were submitted in NCBI Sequence Read Archive (SRA) under the Bioproject ID PRJNA543007. As the NCBI data is not available publicly before publication you can download all fastq files from NCBI reviewer link- <https://dataview.ncbi.nlm.nih.gov/object/PRJNA543007?reviewer=6786803bb25a21ovsuasnddc>

Q14: Aside from these concerns, I would like to thank the authors for their ambitious data analyses. This research provides an important touchstone at a critical time for cultivation techniques in the clinical setting.

A14: Thank you for this comment and we fully agree on the importance of the study

Q15: Line 16: Do we have references for “many studies have argued for them to replace TCMs”? This is stated in abstract but not explored in the scientific justification. Perhaps this position is advocated in the papers cited regarding 16s rRNA vs. TCM (5, 9-11)?

A15: It is correct that we do not directly repeat this statement in the main section of the manuscript. In the main section, we give a more detailed commentary on the many studies that have explored this question and present their own case where NGS will give a better result than TCM.

Q16: Line 67-69: Are the same children sampled multiple times (1 week, then 1 month, etc) or is each sample from a unique individual? This may be irrelevant, but might be helpful to know if these samples were meant to be understood as time-series or independently?

A16: These are the same children sampled multiple times (1 week, then 1 month, then 3 months for hypopharyngeal samples and 1 week, then 1 month and then 1 year for fecal samples). In this study, we have not considered all samples individually and neither our analysis or discussion considers the samples as time series data. In our opinion, including an analysis of the development within each child is outside the scope of our study and would make the conclusions of the study less clear.

Q17: Line 38: “Molecular diagnostic techniques, such as Polymerase chain reaction (PCR), DNA fingerprinting and Next-Generation Sequencing (NGS) may be a game changer” seems a bit vague for a lead sentence. As the thrust of the article relates to diagnostic capacity of 16s rRNA are there any citations that could be applied here to support this claim of being a game-changer?

A17: We thank the reviewer for bringing this up. We have now changed the two first sentences of the paragraph to read- “Molecular diagnostic techniques, such as Polymerase chain reaction (PCR), DNA fingerprinting and Next-Generation Sequencing (NGS) may have been come a game changer. These techniques have now become increasingly rapid, sensitive and cost-efficient”. Furthermore, we have now included 2 references that assess how such molecular diagnostic techniques have been used to improve understanding and treatment outcome in clinical care of chronic wounds.(S Dowd et al 2008, PMID: 18325110 and RD Wolcott et al 2010, PMID: 20616768)

Q18: Line 78/161: Is 4% species-resolution when using amplicon sequence variants (ASV) typical? If this is comparable to other studies a citation would be helpful, if not an explanation for possible sequencing error/contamination might also be needed.

A18: We thank the reviewer for raising a very good point. As this study is itself unique where we have compared the culture vs amplicon sequencing based on ASVs. At the time of writing this manuscript, we did not see any article that talks about species-resolution when using ASVs. The two factors that we expect to be the possible explanations are that some bacterial species are not represented in databases, while for cases where multiple species have exactly the same v4 region; they will only be assigned the lowest common ancestor. The latter point is what we are showing in the section “Amplicon sequencing resolution” (line 155-166).

Due to this, we could not classify the ASVs at the species level. We have now described this in the manuscript for better clarity in line 167-179.

Q19: Line 102/258: Authors should state that RDP stands for Ribosomal Database Project.

A19: Thanks for your kind suggestions. We have revised the manuscript to change “RDP” to “Ribosomal Database Project (RDP)” in line 104

Q20: Line 125-127: Is it possible to clarify this statement: “When considering a presence/absence scenario among the 106 species, bacteria identified by both culturing and sequencing represented 75.70 % of all cultured bacteria, but only accounted for 23.86 % of

sequenced bacteria “ Is this to say: Of the 106 identified unique species from culture, 75.7% could also be recovered in sequencing reads with 16s rRNA methodology. While these 106 culturable species represented just 23.86% of the unique bacteria found via 16s rRNA?

A20: We have been fighting this sentence for a very long time to express exactly our results. We have now updated the entire paragraph to say- “When considering a presence/absence scenario for the 106 species, 75.7% of the times a bacteria was found and identified by culturing, amplicon sequencing identified the same bacteria from the same sample. Likewise, counting each time one of the 106 bacteria was identified in a sample, culturing had identified the same bacteria in the same sample 23.86% of the times (Supplementary Table 7).”

Additional changes made by the authors

Line 20: added word “methods”

Line 33: added word “pathogenic”

Line 40: we have also revised the following text in the introduction “may become a” to “have been”

Line 41: deleted the word “now”

Line 58: changed the word “shown” to “performed”

Line 59: removed word “mostly”

Line 61: added word “in diagnostics”

Line 88-90: we have revised the family abundance at the second digit.

Line 106: changed number from “20” to “22”

Line 108: added Supplementary table 4.

Line 109: changed number from “41” to “40”

Line 159: added word “the species”

Line 161: added line “based on the sequence of their V4 region of the 16S rRNA gene”

Line 165: changed word “From” to “For”

Line 207: deleted the word “anaerobes”

Line 208 deleted the word “gut”

Line 226: replaced “similarly” with “while”

Line 227: replaced “failed to detect” with “not detected”

Line 241: added word “for”

Line 244-245: Line added, “even if TCM were to be replaced for clinical testing,”

Table 1: “Corynebacterium sp” changed to “Corynebacterium spp”

Table 2: “Sequencing” changed to “Amplicon sequencing”

Supplementary tables: We have further revised the manuscript from sequencing to amplicon sequencing in the supplementary table 2, 3, 6, 7, and 8.

We have further revised the Supplementary table 4 to “Complete list of species identified by culturing and their percentage and which unique bacterial species they belong to.”

REVIEWERS' COMMENTS:

Reviewer #1 (Remarks to the Author):

The authors resolved all of my questions.

Two minor errors :

1. Supplementary Table 8: Comparison of the 106 unique bacteria detected identified. The detected and identified are a duplicate.
2. For supplementary figure 3, why the 1-week rarefaction curve (Yellow one) show a decreasing trend as sequencing depth increase? Is there any error here while producing this figure?

Reviewer #2 (Remarks to the Author):

The revisions greatly improve the paper and I'm excited to see this paper carry forward in the editorial process.

line 14 - what is your support for this statement?

"Next-Generation Sequencing (NGS) of 16S rRNA gene is now the most widely used application to investigate the microbiota at any given body site in research."

line 38 - "Molecular diagnostic techniques, such as Polymerase chain reaction (PCR), DNA fingerprinting and Next-Generation Sequencing (NGS) have been game changer^{2,6}." should probably either read "have been game changers" or "have been a game changer".

the authors use the term "antibiotic resistance" (lines 33, 53) but do not speak of "antimicrobial susceptibility testing", which is truly what TCM methods allow in the clinical setting, a subtle piece of vernacular but important when talking about individual patient care.

lines 155-178 and updates to discussion are much appreciated. Though I am sympathetic to the authors' great work and message here, the limits of 16S-NGS are pretty profound in clinical practice (no antimicrobial susceptibility testing, no virulence information, poor resolution of speciation, high cost (especially high marginal costs), high hands-on time, reporting decisions and clinical interpretation of 140 species per specimen (clinical labs would probably not report >1 of the species recovered by TCM or NGS in this study...) -- not to mention the sunk costs of >100 years of clinical correlates of culture data, which may take considerable time for NGS to obtain...), and why clinical laboratories are still currently investing in total laboratory automation for culture plates and scaling culture practices, despite the limited sensitivity the authors note here. The revisions do a better job of explaining these limitations, beyond the main top-line finding that many things are missed by culture.

Discussion -- Also, the authors need to acknowledge that looking at fecal samples and hypopharyngeal aspirations in a prospective cohort study of asthma has limitations as a cohort to look at TCM versus NGS, since it does not seem as though the children had any clinical symptoms at the time they were evaluated, no?

Reviewer #3 (Remarks to the Author):

No further comments to add at this time. Thank you to the authors for addressing the comments/concerns from the initial review.

Revision 2nd

Reviewer comments

Author's comments

Reviewer #1 (Remarks to the Author):

The authors resolved all of my questions.

Two minor errors:

Q1. Supplementary Table 8: Comparison of the 106 unique bacteria detected identified. The detected and identified are a duplicate.

A1. Thank you for reviewing the supplementary material thoroughly enough to catch this mistake. We have now removed "detected" so the text reads, "Comparison of the 106 unique bacteria identified"

Q2. For supplementary figure 3, why the 1-week rarefaction curve (Yellow one) show a decreasing trend as sequencing depth increase? Is there any error here while producing this figure?

A2. We are aware that a rarefaction curve for a sample cannot decrease. We have calculated the mean diversity for all samples from each time point. As the diversity is not calculated beyond the actual depth of each sample, this means that the curve can dip when extending past the depth of samples that have been above mean before then.

Reviewer #2 (Remarks to the Author):

The revisions greatly improve the paper and I'm excited to see this paper carry forward in the editorial process.

Q4. line 14 - what is your support for this statement?

"Next-Generation Sequencing (NGS) of 16S rRNA gene is now the most widely used application to investigate the microbiota at any given body site in research."

A4. We believe 16S rRNA gene amplicon sequencing is the most common technique used to study the human microbiome. This is the case due to the rapid and cheap costs of each run nowadays. There are thousands of papers now published each year linking specific microbes and/or host-microbe co-metabolites to specific diseases, physiological properties, or environmental parameters; and many papers argue this point. However, in a field that is constantly evolving, researchers are now indeed moving towards metagenomics. Therefore, with no direct comparative study statistically backing this claim, we have now changed the sentence to read the following: "Next-Generation Sequencing (NGS) of 16S rRNA gene is now **one of** the most widely used application to investigate the microbiota at any given body site in research" as we think this is an acceptable statement to make without reference or analytical support

Q5. line 38 - "Molecular diagnostic techniques, such as Polymerase chain reaction (PCR), DNA fingerprinting and Next-Generation Sequencing (NGS) have been game changer^{2,6}." should probably either read "have been game changers" or "have been a game changer".

A5. Thanks for your kind suggestion. We have revised the manuscript to change "have been game changer" to "have been game changers" in line 38.

Q6. the authors use the term "antibiotic resistance" (lines 33, 53) but do not speak of "antimicrobial susceptibility testing", which is truly what TCM methods allow in the clinical setting, a subtle piece of vernacular but important when talking about individual patient care.

A6. Thank you for raising this point. In line 33 we have changed "evaluate antibiotic resistance" to "antimicrobial susceptibility testing" and in line 53 we have changed "antibiotic resistance" to "antimicrobial susceptibility"

Q7. lines 155-178 and updates to discussion are much appreciated. Though I am sympathetic to the authors' great work and message here, the limits of 16S-NGS are pretty profound in clinical practice (no antimicrobial susceptibility testing, no virulence information, poor resolution of speciation, high cost (especially high marginal costs), high hands-on time, reporting decisions and clinical interpretation of 140 species per specimen (clinical labs would probably not report >1 of the species recovered by TCM or NGS in this study...) -- not to mention the sunk costs of >100 years of clinical correlates of culture data, which may take considerable time for NGS to obtain...), and why clinical laboratories are still currently investing in total laboratory automation for culture plates and scaling culture practices, despite the limited sensitivity the authors note here. The revisions do a better job of explaining these limitations, beyond the main top-line finding that many things are missed by culture.

A7. We think that this point is very relevant and we agree that this part of the manuscript have been noticeably improved by the reviewer's earlier comments.

Q8. Discussion -- Also, the authors need to acknowledge that looking at fecal samples and hypopharyngeal aspirations in a prospective cohort study of asthma has limitations as a cohort to look at TCM versus NGS, since it does not seem as though the children had any clinical symptoms at the time they were evaluated, no?

A8. Yes, the reviewer is correct in the assessment that our study was performed using samples from healthy children that were part of a large cohort study. However, the use of molecular diagnostics can still be justified when working with samples from infected patients; its applications are not limited to samples acquired from healthy patients. For example, one of the references we cite in this manuscript Wolcott *et al.* 2010 compared healing outcomes at their wound healing center both before and after the introduction of molecular diagnostics¹. Their results indicated that molecular diagnostic applications allow comprehensive evaluation of the microbial bioburden in chronic wounds and in turn, they were able to implement a more precise and targeted therapeutic approach to chronic wound care.

¹Wolcott, R. D., Cox, S. B. & Dowd, S. E. Healing and healing rates of chronic wounds in the age of molecular pathogen diagnostics. *J. Wound Care* 19, 276–284 (2010).

Reviewer #3 (Remarks to the Author):

Q9. No further comments to add at this time. Thank you to the authors for addressing the comments/concerns from the initial review.

A9. Thanks for encouraging remarks and kind suggestion to improve the manuscript for publication.

Additional changes made by the authors

Revised the abstract according to the Journal guidelines, by keeping it under 150 words length.

Change “Main” heading to “Introduction” from line 24

Revised the final paragraph of Introduction by adding the major results and conclusions, as suggested by the editors

Removed word “significantly” from line 198.

Deleted inverted commas from 286, 300

Change number from “26.50 %” to “27.63 %” in line 133

Revision 1st

Reviewer comments

Author’s comments

Reviewer #1

This manuscript touches a very interesting topic of whether 16S rRNA amplicon sequencing is ready or not to replace the culturing in clinical diagnosis using a relatively big sample size. The methodology is nothing novel and the discoveries are highly expected that sequencing will obtain much more species than culturing. However, the comparisons still have their value in terms of comparing these two technologies in the clinical application scenario. It is good to have some big sample size data to show how the two technologies differ in terms of identifying the bacterial species. Hence, I think it holds the value to be published.

Q1: The detailed table of which bacteria species are cultured from which sample should be listed in the supplementary tables in detail. The table is the resource and part of the major value of this study.

A1: Thanks for your kind suggestions. We have now included a link to the data in the data availability section.

Q2: Due to the mature of NGS and the popularity of third generation sequencing (Pacbio and Nanopore), full-length 16S rRNA sequencing will be much more accurate to achieve species-level identification. And these technologies may have more possibility to replace culturing rather than sequencing a hypervariable region of 16S rRNA discussed in this manuscript. Moreover, for high biomass samples like faecal samples, shotgun metagenomics can achieve a higher resolution to strain level can be used. In considering this the method used in this study cannot represent sequencing. The scope of the title culturing vs sequencing is not accurate, it is more accurate of culturing vs 16S rRNA hypervariable sequencing. This part should be mentioned in the discussion.

A2: We agree with the reviewer therefore we changed the title to “Culture vs amplicon sequencing”. We have further revised the manuscript to amplicon sequencing (line 55, 65, 102, 129, 155, 156, 194, 214, 226 and 277).

Q3: For data sharing, the authors should deposit the raw sequences data to the public database and also the metadata associated with the samples.

A3: We apologise for not having made the data available before submission. We have now included a data availability section and made both sequencing and culturing data available. The sequences were submitted in NCBI Sequence Read Archive (SRA) under the Bioproject ID PRJNA543007. As the NCBI data is not available publicly before publication you can download all fastq files from NCBI reviewer link-

[https://dataview.ncbi.nlm.nih.gov/object/PRJNA543007?reviewer=6786803bb25a21ovsuasnddc](https://dataview.ncbi.nlm.nih.gov/object/PRJNA543007?reviewer=6786803bb25a21ovsuasnddc&view=table)

Reviewer #2 (Remarks to the Author):

This study compares V4 sequencing versus traditional culture in a large cohort. The size of the study is intriguing, but the article is confusingly written and does not simply lay out methods and findings. The number of reads per specimen is incredibly low. The article could be substantially improved by including more descriptive legends on the Figures. More description is needed on how TCM was conducted as the comparator method especially in terms of how the bacterial isolates were chosen for Vitek characterization.

Q4: line 21-22 - unclear whether these percentages describe the percentage of all species of bacteria detected or is the percent abundance covered. Also need to say at what taxonomical category this was covered.

A4: Thank you for bringing this to our attention. The percentage described here refers to the average percentage of unique bacterial species identified in a given sample by the two methods (abundances not considered).

Q5: line 79 - why rarefy the samples to so few reads per sample?

A5: We agree that it would have been better if we could include more reads, but our analysis is sensitive to sequencing depth, which makes rarefaction unavoidable. Based on the rarefaction curves we chose to exclude samples with less than 2000 reads and then, to avoid bias of

having unrarefied samples we rarefied to 90% of the reads in the sample with the least reads (2006 reads).

Despite the level of reads per sample, we reach a resolution where one read represents less than 0.06% of the total abundance, and we are confident in our results as any increased sequencing depth can be expected to have amplified the results.

Q6: more discussion of the TCM is required. was every cultured bacteria speciated? usually this is not done for fecal culture.

A6: We are not certain we understand this query, does the reviewer wish that we expand the TCM part in the method section or in the result section?

While it might not be how others have done this, our aim was to identify as many bacteria as possible using the standard media and incubation conditions.

We have updated line 270-276 in the method section so that they now read- "Subsequently, based on the growth on selective media, characteristics of colonies, and cellular morphology, all unique bacterial colonies were isolated. All bacterial isolates were identified biochemically by the automated identification system VITEK-2 (BioMérieux, France). Bacteria cultured anaerobically were not identified further. All isolates were preserved at -80°C. No quantification was performed". Furthermore, we added a missed reference to Stokholm et al 2011 where the isolates from the cohort's vaginal samples were presented.

Q7: There needs to be more discussion as to why NGS would miss any of the bacteria detected by culture.

A7: We have been surprised by this as well and decided to look at the data again. Of the 110 times *H. influenzae* was identified by TCM sequencing identified *H. haemolyticus* 92 times (which was not cultured in those samples). As the difference between the sequence of the two bacteria is just 2/253 bp, this indicates that the bioinformatic pipeline cannot correctly differentiate between the two species. We have made this comparison for the 5 species mentioned in line 136-140, and added this as an extra paragraph in the results section for amplicon sequencing resolution (line 167-179).

Additionally, we have also added the following text to the discussion (line 231-237):

"Additionally, our results show that amplicon sequencing had some problems with correct separation of sequences that differ in as little as 1-2 bp out of 253 bp. An important step in the process is to create an error model that can infer if a single nucleotide difference is a sequencing error or an actual sequence variant, and it is possible that with the high sequence similarity the error model wrongly classifies the differences as a sequencing error and then changes the sequences to match the closest matching sequence."

Q8: Figure 1 - unclear from legend what the time period is covered. Unclear what unit the Abundance is in.

A8: To improve the clarity of the figure legend we have changed it to- "Boxplot showing the mean relative abundance of bacteria classified using ASVs, matching bacteria identified, by

culturing, in each sample. The analysis was performed for fecal (left panel) and hypopharyngeal (right panel) samples separately and at all taxonomic levels from Phylum to Species (x-axis) and the color of each box denotes the timepoint.”

Q9: Figure 2 - this figure is totally unclear to me. It is not clear what the denominator is on the Y axis. The legend is not written in complete sentences. Why not just say "Culture detected XX% of the total number of species found by sequence in XX sample".

A9: To improve the clarity of the figure legend we have changed it to- “Fig. 2: Barplot showing the percentage of bacteria identified by amplicon sequencing that were also identified by TCM in the same sample. The three groups of bars (x-axis) show the result for the different sample types (all together or fecal and hypopharyngeal samples separately). Blue bars show the result for dominant bacteria (abundance > 10%), orange show major bacteria (1-10%), and grey show minor bacteria (< 1%), while the yellow line indicates the overall percentage for each sample type.”

Reviewer #3 (Remarks to the Author):

The study by Gupta et. al compares the diagnostic efficiency of traditional culture methods and 16s rRNA sequencing using a large number of fecal and hypopharyngeal samples in healthy infants. The study is remarkable for its design, which pairs each methodology in each metagenome and in its scope, analyzing over 3,500 samples. This investigation highlights the strengths and weaknesses of each approach conveying the influence of sampling site, subject age, and microbial abundance within a sample on the efficiency of each methodology. The study is well-written and organized in a way that lends to understanding its key concepts. Overall, this investigation provides an important frame of reference for those hoping to understand the link between longstanding cultivation methods and a future shifting toward clinical applications of next generation sequencing.

Q10: One aspect of the paper which does not seem fully settled is whether or not the authors have reached a definite conclusion on the central question: “Are we ready for this transition?” While the authors conclude that with development of a “16s rRNA gene amplicon sequencing workflow” and “optimized oligonucleotide arrays for detecting various genes” we might be ready for replacement of traditional culture methods, they end by stating that further refinement of techniques “involving bacterial behavior and interactions” is needed. This conclusion feels somewhat vague in the context of the extensive analysis they have undertaken.

A10: At this point we cannot conclude that we are ready for a transition away from traditional culture methods for clinical work. We believe that our results, together with the latest developments in sequencing technology, shows that the technological tools are available to make the transition. Unfortunately, we need a polished protocol/package that can rapidly handle, interpret and present sequencing result before this transition can be made. While we would prefer to provide a very strong conclusion about our results we find that this is not supported by the results presented in the manuscript.

The point we want to make with the last sentence is that while traditional culture methods might disappear from clinical laboratories, the techniques can still be improved as they are still highly relevant for research involving bacterial behavior and interactions. We have expanded the last sentence (line 244-245) so it now reads: "However, even if TCM were to be replaced for clinical testing, there is still a need for further refining culture techniques for research involving bacterial behavior and interactions. "

Q11: For instance, the authors make an effort to establish that resolution to the species level is often not possible with sequencing of 16s rRNA gene amplicons (especially in cases when only the v4 region is analysed). They also point out that in this study some potentially pathogenic organisms, such as haemophilus influenzae, were cultured only via traditional methods and not recovered from sequencing reads in a large number of metagenomes, a finding which in the right clinic setting is of huge significance.

A11: This point is very relevant and was also raised by reviewer number 2. Therefore, we will refer to our answer to query number 7.

Q12: Additionally, the authors make the claim that TCMs are biased toward relative abundance of a microbial population within a metagenome and they claim that this "may be a good thing in a clinical setting, where you have no intention on initializing treatment on a bacteria that is not infecting but only colonizes". However, the importance of these findings does not seem directly linked to a strong conclusion about the current state of methodology in clinical application.

A11: We agree that the references section is not linked to a strong conclusion, but this part is more speculative and we are of the opinion that our data do provide the necessary support to make a strong conclusion about the current state of methodology in clinical applications.

Q12: One question that may linger in the minds of readers is how 16s rRNA compares to other methods of high-throughput sequencing. In particular, shotgun metagenomics might overcome the lack of species/strain resolution seen with 16s rRNA and would allow researchers to more fully understand the behaviour and interactions within the metagenomes being studied. At the same time shotgun data is complex, data analysis is time consuming and the cost per sample is much higher than 16s rRNA. While these limitations may make shotgun metagenomics less relevant for imminent clinical application, a brief reflection on this modality may help orient the reader.

A12: Thanks for your kind suggestions. While we fully agree with this query, and have considered to discuss this in the manuscript, we felt that this would be a discussion for another article where the methods could be compared as well. Furthermore, as the reviewer mentions in the end, shotgun sequencing is less relevant for imminent clinical application.

In response to reviewer 1 (Q2), bringing up that we did not clearly state the type of sequencing used in this study, we have changed the title of the manuscript to "Culture vs amplicon sequencing" for clarity. Furthermore, we have further revised the manuscript to amplicon sequencing (line 55, 65, 102, 129, 155, 156, 194, 214, 226 and 277).

Q13: As best I am able to tell from reviewing the manuscript it does not seem the data generated in this study has been made available with a public database. In general, data

availability is essential for credibility and allows proper scrutiny of analyses by the reviewers and the community.

A:13: We apologise for not having made the data available before submission. We have now included a data availability section and made both sequencing and culturing data available. The sequences were submitted in NCBI Sequence Read Archive (SRA) under the Bioproject ID PRJNA543007. As the NCBI data is not available publicly before publication you can download all fastq files from NCBI reviewer link-

<https://dataview.ncbi.nlm.nih.gov/object/PRJNA543007?reviewer=67868o3bb25a21ovsuasnddc>

Q14: Aside from these concerns, I would like to thank the authors for their ambitious data analyses. This research provides an important touchstone at a critical time for cultivation techniques in the clinical setting.

A14: Thank you for this comment and we fully agree on the importance of the study

Q15: Line 16: Do we have references for “many studies have argued for them to replace TCMs”? This is stated in abstract but not explored in the scientific justification. Perhaps this position is advocated in the papers cited regarding 16s rRNA vs. TCM (5, 9-11)?

A15: It is correct that we do not directly repeat this statement in the main section of the manuscript. In the main section, we give a more detailed commentary on the many studies that have explored this question and present their own case where NGS will give a better result than TCM.

Q16: Line 67-69: Are the same children sampled multiple times (1 week, then 1 month, etc) or is each sample from a unique individual? This may be irrelevant, but might be helpful to know if these samples were meant to be understood as time-series or independently?

A16: These are the same children sampled multiple times (1 week, then 1 month, then 3 months for hypopharyngeal samples and 1 week, then 1 month and then 1 year for fecal samples). In this study, we have not considered all samples individually and neither our analysis or discussion considers the samples as time series data. In our opinion, including an analysis of the development within each child is outside the scope of our study and would make the conclusions of the study less clear.

Q17: Line 38: “Molecular diagnostic techniques, such as Polymerase chain reaction (PCR), DNA fingerprinting and Next-Generation Sequencing (NGS) may be a game changer” seems a bit vague for a lead sentence. As the thrust of the article relates to diagnostic capacity of 16s rRNA are there any citations that could be applied here to support this claim of being a game-changer?

A17: We thank the reviewer for bringing this up. We have now changed the two first sentences of the paragraph to read- “Molecular diagnostic techniques, such as Polymerase chain reaction (PCR), DNA fingerprinting and Next-Generation Sequencing (NGS) may have become a game changer. These techniques have now become increasingly rapid, sensitive and cost-efficient”. Furthermore, we have now included 2 references that assess how such molecular

diagnostic techniques have been used to improve understanding and treatment outcome in clinical care of chronic wounds.(S Dowd et al 2008, PMID: 18325110 and RD Wolcott et al 2010, PMID: 20616768)

Q18: Line 78/161: Is 4% species-resolution when using amplicon sequence variants (ASV) typical? If this is comparable to other studies a citation would be helpful, if not an explanation for possible sequencing error/contamination might also be needed.

A18: We thank the reviewer for raising a very good point. As this study is itself unique where we have compared the culture vs amplicon sequencing based on ASVs. At the time of writing this manuscript, we did not see any article that talks about species-resolution when using ASVs. The two factors that we expect to be the possible explanations are that some bacterial species are not represented in databases, while for cases where multiple species have exactly the same v4 region; they will only be assigned the lowest common ancestor. The latter point is what we are showing in the section “Amplicon sequencing resolution” (line 155-166).

Due to this, we could not classify the ASVs at the species level. We have now described this in the manuscript for better clarity in line 167-179.

Q19: Line 102/258: Authors should state that RDP stands for Ribosomal Database Project.

A19: Thanks for your kind suggestions. We have revised the manuscript to change “RDP” to “Ribosomal Database Project (RDP)” in line 104

Q20: Line 125-127: Is it possible to clarify this statement: “When considering a presence/absence scenario among the 106 species, bacteria identified by both culturing and sequencing represented 75.70 % of all cultured bacteria, but only accounted for 23.86 % of sequenced bacteria “ Is this to say: Of the 106 identified unique species from culture, 75.7% could also be recovered in sequencing reads with 16s rRNA methodology. While these 106 culturable species represented just 23.86% of the unique bacteria found via 16s rRNA?

A20: We have been fighting this sentence for a very long time to express exactly our results. We have now updated the entire paragraph to say- “When considering a presence/absence scenario for the 106 species, 75.7% of the times a bacteria was found and identified by culturing, amplicon sequencing identified the same bacteria from the same sample. Likewise, counting each time one of the 106 bacteria was identified in a sample, culturing had identified the same bacteria in the same sample 23.86% of the times (Supplementary Table 7).”

Additional changes made by the authors

Line 20: added word “methods”

Line 33: added word “pathogenic”

Line 40: we have also revised the following text in the introduction “may become a” to “have been”

Line 41: deleted the word “now”

Line 58: changed the word “shown” to “performed”

Line 59: removed word “mostly”

Line 61: added word “in diagnostics”

Line 88-90: we have revised the family abundance at the second digit.

Line 106: changed number from “20” to “22”

Line 108: added Supplementary table 4.

Line 109: changed number from “41” to “40”

Line 159: added word “the species”

Line 161: added line “based on the sequence of their V4 region of the 16S rRNA gene”

Line 165: changed word “From” to “For”

Line 207: deleted the word “anaerobes”

Line 208 deleted the word “gut”

Line 226: replaced “similarly” with “while”

Line 227: replaced “failed to detect” with “not detected”

Line 241: added word “for”

Line 244-245: Line added, “even if TCM were to be replaced for clinical testing,”

Table 1: “Corynebacterium sp” changed to “Corynebacterium spp”

Table 2: “Sequencing” changed to “Amplicon sequencing”

Supplementary tables: We have further revised the manuscript from sequencing to amplicon sequencing in the supplementary table 2, 3, 6, 7, and 8.

We have further revised the Supplementary table 4 to “Complete list of species identified by culturing and their percentage and which unique bacterial species they belong to.”